# TrafficBT: Advancing Pre-trained Language Models for Network Traffic Classification with Multimodal Traffic Representations

## Abstract

Advances in pre-training and large language models have led to the widespread adoption of pre-trained models for network traffic classification, enhancing service quality, security, and stability. However, most existing pre-trained methods focus solely on payload semantics, neglect temporal dependencies between packets, and rely on single-dimensional static feature learning. This limitation reduces their robustness and generalization capabilities in dynamic and heterogeneous network environments. To address these challenges, we propose `TrafficBT`, a universal traffic classification framework combining pre-training with multimodal fine-tuning. It extracts both semantic and spatio-temporal features and uses data augmentation to handle data scarcity and class imbalance. During pre-training, `TrafficBT` leverages large-scale public and real-world traffic datasets to learn domain-specific semantic representations from payloads. In the fine-tuning stage, it adopts a multimodal learning framework that employs a gating network to fuse BERT with a three-layer Transformer architecture, enabling the model to effectively capture both payload semantics and temporal transmission patterns. Experiments show that `TrafficBT` achieves F1 scores above 0.99 on most real-world and benchmark datasets and outperforms eight state-of-the-art baselines across eight downstream tasks. Notably, it improves performance by 21% in encrypted proxy website classification, demonstrating strong robustness and generalization.

## 1 Introduction

With the advancement of network technologies and growing concerns, encrypted traffic has become increasingly prevalent in network communications (Wan et al., 2025). It is accompanied by complex transmission patterns and protocol stru ctures. Real-world encrypted traffic classification tasks, such as VPN detection and malware analysis (Zhao et al., 2023; Zhang et al., 2025), are vital for cybersecurity enforcement, while encrypted application classification and tunnel website identification support traffic visibility and control (Lin et al., 2022). Therefore, analyzing diverse encrypted traffic efficiently and accurately has become a key challenge in modern network environments.

Network traffic classification methods can be categorized into traditional Deep Packet Inspection (DPI) (Sherry et al., 2015), fingerprinting (Crotti et al., 2007), machine learning (Pacheco et al., 2018), deep learning (Rezaei & Liu, 2019; Qiu et al., 2025), and pre-trained model-based technique (Lin et al., 2022). With the rise of encrypted traffic, the reliance on DPI on plaintext limits its effectiveness. Fingerprinting and learning-based methods avoid the reliance on plaintext but often depend on handcrafted features, require large labeled datasets, and struggle with robustness and generalization under diverse network conditions.

Pre-training techniques mitigate labeled data scarcity and poor generalization via two stages, *i.e.*, pre-training and fine-tuning. Self-supervised pre-training on large unlabeled datasets enables the model to capture domain knowledge, and fine-tuning adapts the model to specific downstream tasks (Tang et al., 2022). This paradigm, successful in computer vision and natural language processing, has recently been applied to network traffic classification. Representative models such as ET-BER (Lin et al., 2022), TrafficFormer (Zhou et al., 2025), NetMamba (Wang et al., 2024), and YaTC (Zhao

et al., 2023) outperform traditional methods with stronger feature representations, higher accuracy, and better adaptability for encrypted traffic classification.

However, existing pre-trained traffic models predominantly focus on payload semantics, neglecting crucial temporal features and transmission patterns essential for characterizing traffic behavior. This limitation becomes critical when analyzing encrypted traffic, where inaccessible content makes transmission patterns the key distinguishing factor (Chen et al., 2025). Single-feature learning models often fail to capture these dynamic network behaviors, leading to limited generalization. Furthermore, due to undersampling rare classes, most of the existing methods degrade feature learning and recognition performance. These shortcomings highlight the need for more robust approaches.

To address the above challenges, we propose `TrafficBT`, a general network traffic classification framework that combines pre-training with multimodal fine-tuning. First, to address the severe class imbalance and sample scarcity in public and real-world datasets, we design modality-specific data augmentation strategies for BERT and a lightweight Transformer encoder in `TrafficBT`, enhancing robustness to rare classes and improving overall performance. Second, we pre-trained a BERT(Devlin et al., 2019) model on large-scale real-world and public datasets to learn network traffic payload features. We then design TriFormer, a 3-layer Transformer model for multimodal fine-tuning, which effectively learns dynamic spatio-temporal representations from both flow-level and packet-level statistics. Furthermore, we employ a gated network to fuse dynamic spatio-temporal representations from TriFormer with static payload semantics from BERT, which enables the fine-tuned model to capture features from multiple modalities, enhancing robustness and generalization across downstream tasks. Finally, we evaluate `TrafficBT` on fifteen benchmark datasets covering eight typical downstream tasks. The main contributions of this work are summarized as follows:

- We propose `TrafficBT`, a universal network traffic classification framework that integrates pre-training and multimodal fine-tuning to capture both payload semantics and temporal transmission patterns, addressing the limitations of existing methods.
- We propose a novel multimodal learning mechanism using a gated network that fuses dynamic spatio-temporal representations from TriFormer with static payload semantics from BERT, enabling robust feature extraction across diverse network scenarios.
- We conduct extensive experiments on fifteen public datasets involving eight typical downstream tasks. The results show that `TrafficBT` outperforms eight state-of-the-art baselines and achieves the best performance. In particular, on the website classification task under the encrypted proxy task, it exceeds the best baseline by 21%, demonstrating its superior robustness and generalization ability.

## 2 RELATED WORK

### 2.1 TRADITIONAL TRAFFIC CLASSIFICATION METHODS

**Rule-based Methods.** These methods classify traffic using predefined rules, protocol specifications, or signature libraries, such as DPI and fingerprint matching (Sherry et al., 2015; Crotti et al., 2007). Despite their interpretability, they perform poorly on encrypted traffic. For example, Flowprint (Van Ede et al., 2020) constructs fingerprints from unencrypted packet protocol fields, working well in lightly encrypted scenarios but degrading under complex or end-to-end encryption due to its reliance on plaintext features.

**Machine Learning-based Methods.** These methods leverage flow-level statistics (*e.g.*, packet size, inter-arrival time) to train classifiers such as Decision Trees, Random Forests, Support Vector Machines (SVM), and K-Nearest Neighbors (KNN). By focusing on flow patterns rather than payloads, they exhibit greater resilience to encryption compared to rule-based approaches. However, these methods heavily rely on manual feature engineering. Representative methods like Appscanner (Taylor et al., 2016) use Random Forest classifiers with 54 handcrafted statistical features. Although these methods can be effective in certain scenarios, their applicability may be limited under dynamic traffic patterns or strong encryption.

**Deep Learning-based Methods.** Deep learning methods automatically extract features from raw traffic data for end-to-end classification. Common models include Convolutional Neural Networks

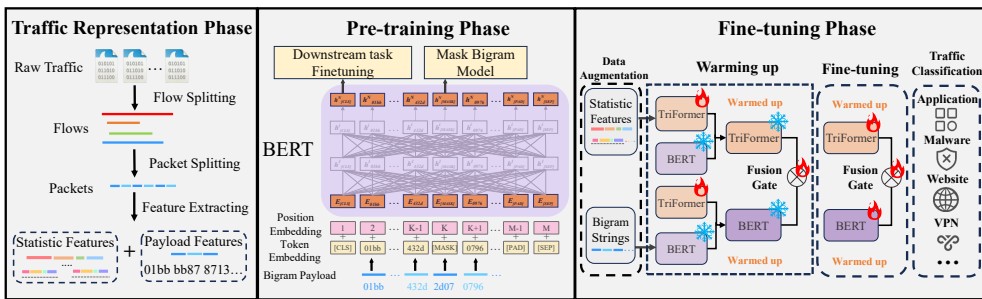

Figure 1: The schematic illustration of the `TrafficBT` Framework.

(CNNs) for raw bytes or images, Recurrent Neural Networks (RNNs) for temporal patterns, Graph Neural Networks (GNNs) for flow-structured graphs, and Transformers for capturing long-range dependencies. They outperform traditional methods and handle encrypted traffic well, but often lack generalizability across tasks. Representative models include FS-Ne (Liu et al., 2019a), which uses RNNs on packet length sequences, and GraphDApp (Shen et al., 2021) and TFE-GNN (Zhang et al., 2023), which apply GNNs on flow graphs.

### 2.2 PRE-TRAINED MODELS FOR TRAFFIC CLASSIFICATION

**BERT-based Pre-training Methods.** These methods leverage BERT's strength in contextual modeling to capture sequential dependencies within network flows for improved traffic representations. Representative models include ET-BERT (Lin et al., 2022) and TrafficFormer (Zhou et al., 2025), both based on the *BERT-base* architecture with tokenizers trained on hexadecimal payloads to better capture domain-specific characteristics. For example, ET-BERT segments traffic into bursts by 5-tuples and employs masked burst modeling and same-origin burst prediction as self-supervised tasks to capture burst-level semantics, enabling multi-task fine-tuning for diverse downstream tasks.

**MAE-based Pre-training Methods.** These methods transform network traffic into image-like matrices (Hang et al., 2023). They then apply a Masked Autoencoder (MAE) (He et al., 2022) for self-supervised learning to capture latent structural patterns. It improves the detection of hidden anomalies and unknown attacks. YaTC (Zhao et al., 2023) segments traffic into packets and flows, constructs multi-level flow representation matrices as MAE input, and uses masked reconstruction to learn semantic structures. The pre-trained MAE encoder is then fine-tuned with a linear classifier for downstream tasks.

However, the above classic pre-trained methods mainly focus on payload semantics and ignore the multimodal traffic representation (*i.e.*, temporal features and transmission patterns) necessary to characterize traffic behavior. Therefore, in this paper, we focus on the use of multimodal traffic representation to enhance model performance.

## 3 METHODOLOGY

**Key Challenge.** Achieving a general and robust classification of encrypted traffic remains a significant challenge. The key challenge lies in jointly learning payload semantics and spatio-temporal statistical features. Due to input length limitations, existing pre-trained models typically process only three to five concatenated packets. In contrast, a typical network flow comprises dozens of packets. As a result, these pre-trained models struggle to capture temporal dynamics and sequential flow structures while simultaneously modeling payload features. Achieving effective joint learning without changing the original pre-training task remains an open problem.

**Our Solution.** To address the above challenge, we propose `TrafficBT`, a pre-trained language model architecture that leverages multimodal traffic representations, as illustrated in Fig. 1. Specifically, the proposed model comprises three key phases: the traffic representation phase, the pre-tuning phase, and the fine-tuning phase. In the traffic representation phase, raw traffic is segmented

into flows and packets, extracting both spatio-temporal features and payload-level characteristics to capture the inherent semantics of encrypted communication patterns. In the pre-training phase, `TrafficBT` trains BERT on large-scale public and real-world traffic datasets, utilizing data augmentation to mitigate class imbalance and sample scarcity. During the fine-tuning phase, it integrates spatio-temporal traffic features with payload semantics by simultaneously fine-tuning BERT and leveraging the Transformer-based TriFormer module. A gating network then fuses these representations, enabling effective multimodal traffic modeling for downstream tasks.

## 3.1 TRAFFIC REPRESENTATION PHASE

We first segment each `.pcap` file into multiple flows based on the 5-tuple (source/destination IPs and ports and protocol), forming the basic unit for subsequent processing.

**Payload Feature Extraction.** For each flow, we extract the first five packet payloads along with IP and transport-layer headers, convert them to hexadecimal, and concatenate them into a byte sequence. This sequence is split into overlapping bigrams (*e.g.*, 01bb8713 → 01bb, bb87, 8713), each representing a 16-bit token. All possible bigrams form a fixed vocabulary of 65,536 unique tokens, with a direct one-to-one mapping to token IDs. Flows are tokenized with this vocabulary and truncated to 256 tokens. We introduce five special tokens: [CLS] (sequence-level representation), [MASK] (masked modeling), [SEP] (end marker), [PAD] (padding), and [UNK] (out-of-vocab bigrams). The resulting sequences are used for BERT pre-training.

**Spatio-temporal Feature Extraction.** For a given network flow, dynamic spatio-temporal properties are reflected in statistical descriptors capturing its temporal patterns and transmission behavior. To effectively model these aspects, we extract 42 flow-level features and 28 packet-level features, as summarized in Tables 1–2. A comprehensive description of these features is provided in Appendix A.

| Category | Feature Name |
|---|---|
| Time-related | timestamp, delta_time, relative_time, time_since_last_handshake |
| Length & Direction | packet_length, payload_length, direction |
| Last-5 Statistics | avg_delta_time_last_5, uplink_ratio_last_5, avg/std_pkt_len_last_5 |
| Protocol & TCP Flags | protocol_id, tcp_flag_(syn/ack/fin), is_ack_only, seq_diff, window_size |
| TLS-related | tls_record_type, tls_version, cipher_suite_len, handshake_phase, key_update_count |
| Content Statistics | entropy, chi_square, printable_ratio, null_byte_ratio, byte_pair_corr |

Table 1: Statistical Features of Network Traffic Packets.

**Flow & Packet Statistical Features.** Specifically, the flow-level feature extraction scheme encompasses five key dimensions, yielding 42 features that comprehensively describe the structural and behavioral patterns of network flows, as provided in Table 1. We extract packet-level features across six dimensions to capture temporal patterns, protocol behaviors, and content characteristics, as shown in Table 2. Among the above statistical features, categorical and flag-type attributes are encoded into numeric representations using categorical encoding. For numerical features, Min-Max normalization is applied to mitigate scale disparities, particularly for time- and length-related attributes, thereby enhancing training stability and convergence.

**Data Augmentation.** The prevalence of class imbalance in real-world datasets leads to models biased towards majority classes, a problem inadequately addressed by conventional undersampling techniques. We utilize input-specific data augmentation strategies to enhance model robustness and minority class representation. Our augmentations are carefully calibrated to be realistic yet non-disruptive,

| Category | Feature Name |
|---|---|
| Packet Count Statistics | Total Fwd Packets, Total Bwd Packets |
| Packet Length Statistics | Packet Length Min/Max/Mean/Std/Total (Fwd, Bwd, Flow) |
| Inter-Arrival Time (IAT) | IAT Min/Max/Mean/Std/Total (Fwd, Bwd, Flow) |
| TCP Flags | FIN, SYN, RST, PSH, ACK, URG, CWR, ECE Flag Count |
| Flow Rate Statistics | Flow Bytes/s, Flow Packets/s |

Table 2: Statistical Features of Network Traffic Flows.

reflecting that phenomena like packet corruption or reordering seldom exceed a 10% rate in practice. For payload data, we simulate information loss by masking 10% of bytes and enriching structural diversity by shuffling packet order with a 10% probability. For statistical features, we emulate mea-

surement noise by adding 5% local random noise, simulate anomalies by masking 10% of features, and promote temporal dependency learning by shuffling the feature sequence with a 10% probability. Fig. 1 illustrates the overall `TrafficBT` framework integrating these augmentation strategies.

## 3.2 PRE-TRAINING PHASE

**Pre-traing Task.** During pre-training, we omit the Next Sentence Prediction task, as RoBERTa (Liu et al., 2019b) shows that it offers limited performance improvement. Instead, inter-packet pattern learning is deferred to multimodal fine-tuning. The model focuses solely on the Mask Bigram Model task to capture internal payload structures. To this end, we build a large-scale data set of one million network flows, combining real-world mobile app traffic with public VPN, Tor, and DNS datasets. Specifically, 15% of the tokens in each input sequence are randomly selected for masking. Of these, 80% are replaced with `[MASK]`, 10% remain unchanged, and 10% are substituted with random tokens. BERT's bidirectional Transformer architecture leverages both preceding and succeeding contexts to predict masked tokens, effectively capturing the semantic and structural characteristics of the payload. During training, the negative log-likelihood (NLL) loss is utilized to optimize the model and is defined as follows:

$$\mathcal{L}_{\text{MBM}} = -\sum_{i=1}^{k} \log\left(P(\text{MASK}_i = \text{token}_i \mid \tilde{X}; \theta)\right), \tag{1}$$

where $\theta$ denotes the trainable parameters of `TrafficBT`, and $k$ is the number of masked bigram tokens. The conditional probability $P(\cdot)$ is modeled by the Transformer encoder with parameters $\theta$. The input sequence is $\tilde{X}$, $\text{MASK}_i$ is the predicted token at the $i$-th masked position, and $\text{token}_i$ is the original token.

**Layer Freezing Strategy.** After pre-training, we freeze the first 8 BERT layers and keep the top 4 layers trainable for subsequent fine-tuning. This approach reduces trainable parameters and speeds up fine-tuning, beneficial for small datasets or limited resources. Lower layers capture general features and are retained to preserve pre-trained knowledge, while upper layers adapt to specific tasks (Devlin et al., 2019). This strategy also helps prevent overfitting and catastrophic forgetting. Prior studies indicate that freezing intermediate layers often achieves performance comparable to or even better than full fine-tuning, especially in tasks like text classification and question answering (Vilares et al., 2020).

## 3.3 FINE-TUNING PHASE

During the fine-tuning phase, we design TriFormer (see Fig. 2), a dedicated Transformer-based encoder for capturing spatio-temporal statistical features. It is integrated with the pre-trained BERT model via a gating network to enable joint learning of payload semantics and spatio-temporal characteristics. Next, we introduce its architecture in detail.

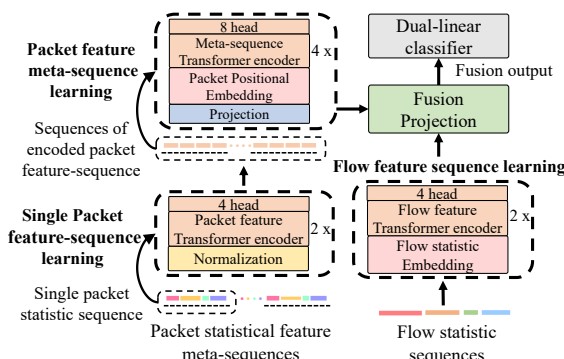

Figure 2: The architecture of the proposed TriFormer.

**TriFormer Architecture.** Fig. 2 illustrates the detailed architecture of Tri-Former, which comprises three hierarchical Transformer encoder modules designed to capture multilevel spatio-temporal patterns from network traffic.

- **Single-Packet Feature Sequence Learning:** A 2-layer Transformer with 4 heads models intrapacket feature sequences, capturing local temporal patterns.
- **Packet Feature Meta-Sequence Learning:** A 4-layer Transformer encoder with 8 attention heads per layer processes the full sequence of packet-level features in a network flow (up to 100 packets). Inputs retain original packet ordering and timestamps to capture critical inter-packet temporal dynamics and transmission patterns.

- **Flow Feature Sequence Learning:** A 2-layer Transformer encoder with 4 attention heads models global flow statistics. This distills high-level flow characteristics to better model cross-packet temporal relationships and multimodal representations.
- **Fusion Projection Layer:** Packet sequences and flow statistics are concatenated and compressed to 256 dimensions via a linear projection, integrating multi-level features for joint modeling with BERT semantics.

**Fusion Gate Design.** The gating network (see Fig. 1) enables adaptive multimodal fusion. Initially, cross-attention between payload semantics and spatio-temporal statistics (inspired by VisualBERT (Li et al., 2019)) with self-attention shows slow convergence and limited gains. Adding residual connections (He et al., 2016) slightly improves convergence. Fusion is then simplified by concatenating features and applying a linear layer, resulting in faster training. Details of these attempts and comparisons are provided in Appendix E.. Finally, we introduce a gating mechanism to learn modality-specific weights and adaptively combine features (Kim et al., 2020). The fused output then undergoes a linear transformation for downstream tasks, which strengthens multimodal representation learning.

For multimodal fusion, we use the [CLS] token from the last hidden layer of BERT (first 8 layers frozen) as the payload semantic representation $\boldsymbol{f}_{\text{BERT}} \in \mathbb{R}^{768}$ (Lin et al., 2022). The encoded output of TriFormer represents the spatio-temporal statistical representation $\boldsymbol{f}_{\text{Tri}} \in \mathbb{R}^{256}$. A linear projection layer reduces $\boldsymbol{f}_{\text{BERT}} \in \mathbb{R}^{768}$ to 256 dimensions to align with $\boldsymbol{f}_{\text{Tri}} \in \mathbb{R}^{256}$. Subsequently, both features are normalized to ensure stable training and avoid gradient issues:

$$\hat{\boldsymbol{f}}_{\text{BERT}} = \text{LayerNorm}(\text{Linear}_{768\rightarrow256}(\boldsymbol{f}_{\text{BERT}})), \quad \hat{\boldsymbol{f}}_{\text{Tri}} = \text{LayerNorm}(\boldsymbol{f}_{\text{Tri}}). \tag{2}$$

We concatenate the normalized features and compute gating weights via three expert branches. The gating coefficient is the maximum of the three weights:

$$\boldsymbol{f}_{\text{cat}} = [\hat{\boldsymbol{f}}_{\text{BERT}}; \hat{\boldsymbol{f}}_{\text{Tri}}] \in \mathbb{R}^{512}, \alpha = \max_{i \in \{1,2,3\}} \sigma(\text{Linear}_i(\boldsymbol{f}_{\text{cat}})) \in (0, 1). \tag{3}$$

Finally, the features are combined via a weighted sum to form a fused representation. This representation is then processed by a four-layer MLP classification head to obtain the final prediction $\hat{y}$:

$$\hat{y} = \text{MLP}(\alpha \cdot \hat{\boldsymbol{f}}_{\text{BERT}} + (1 - \alpha) \cdot \hat{\boldsymbol{f}}_{\text{Tri}}). \tag{4}$$

**Multimodal Fine-tuning Strategy.** The multimodal fine-tuning process (see Fig. 1) adopts a 3-stage warm-up strategy. This design mitigates unstable gradients caused by directly training uninitialized TriFormer and fusion modules with BERT. (1) BERT is warmed up for $n$ epochs, fine-tuning only its last 4 layers while freezing the first 8 layers. To reduce gradient suppression and improve convergence stability, BERT's 1-layer classification head is replaced with a 2-layer head. (2) TriFormer undergoes a warm-up phase of $3n$ epochs through downstream classification training to ensure effective learning of its encoded features. Given that TriFormer converges significantly faster than BERT, simultaneous warm-up accelerates overall training. (3) With both BERT and TriFormer warmed up and their parameters frozen, their hidden representations serve as inputs to the fusion gating network. We then warm up the fusion gate for $2n$ epochs via downstream classification tasks, ensuring a stable and effective fusion learning process during the subsequent full fine-tuning stage.

## 4 EXPERIMENT

### 4.1 EXPERIMENT SETUP

We implement `TrafficBT` with PyTorch 2.1.2 and run all experiments on a single NVIDIA A800 workstation. Then, we provide the details of the pre-training datasets, pre-training settings, fine-tuning datasets, fine-tuning settings, metrics, and baselines.

**Pre-training Datasets.** As shown in Table 3, this study utilizes the NUDT-Mobile (Zhao et al., 2024) dataset, which contains real-world network traffic collected internally by NUDT (China). In addition, publicly available datasets ISCXVPN2016 (Gil et al., 2016), ISCXTor2016 (Lashkari et al., 2017), and CIRA-CIC-DoHBrw-2020 (MontazeriShatoori et al., 2020) are incorporated as supplementary pre-training data sources. A detailed description is in Appendix B.

**Pre-training Settings.** Prior studies indicate that the first 3 to 5 packets of a network flow carry most of the key information (Meng et al., 2023). We extract the first 5 packets (headers and payloads) as BERT input, with a max sequence length of 256, a hidden size of 768, and 12 Transformer layers. Since most flows have fewer than 100 packets, we use the statistical characteristics of the first 100 packets

| Dataset | Size | #Flows | #Classes | Included Protocols |
|---------|------|--------|----------|--------------------|
| NUDT-Mobile | 112.2 GB | 1,157,245 | 280 | TCP, UDP, HTTP, TLSv1.2, SSLv2, WebSocket, ... |
| ISCXVPN2016 | 15.6 GB | 4,824 | 5 | TLSv1.2, SFTP, SSDP, SNMP, NTP, GQUIC, ... |
| ISCXTor2016 | 19.7 GB | 39,018 | 7 | TLSv1.1, TLSv1.2, FTP-DATA, SSL, HTTP, WebSocket, ... |
| CIRA-CIC-DoHBrw-2020 | 75.5 GB | 771,497 | 2 | TCP, TLSv1.2, TLSv1.3, SSLv2, SSL, ... |

Table 3: Overview of Pre-training Datasets.

as input to the TriFormer module. The pre-training phase runs on a single NVIDIA A800 GPU (80 GB memory) with batch size 32 and gradient accumulation over 4 steps (effective batch size 128) for 3 epochs. The initial learning rate is 5e-5, with 1000 warm-up steps and a weight decay of 0.01. Mixed-precision training (FP16) is enabled. The AdamW optimizer and linear learning rate scheduler are used. The training uses HuggingFace Trainer with masked language modeling.

**Fine-tuning Datasets.** To ensure robust evaluation, we evaluate downstream classification on 15 datasets covering eight task types, including VPN and Tor traffic classification, network service and application classification, malware detection, encrypted proxy identification, website classification under proxy, and device classification under attacks. To prevent data overlap, we adopt a leave-one-out strategy, excluding the fine-tuning dataset from pre-training. The 15 datasets span diverse real-world traffic scenarios, including ISCXVPN2016 (Service, APP) (Gil et al., 2016), ISCXTor2016 (Lashkari et al., 2017), USTC-TFC-2016 (Benig, Malware) (Wang et al., 2017), CrossPlatform (Android, iOS) (Ren et al., 2019), NUDT-Mobile (Zhao et al., 2024), Datacon2020 (DataCon Community, 2021a), Datacon2021 (Parts 1,2) (DataCon Community, 2021b), and CIC-IoT 2022 (Flood) (Dadkhah et al., 2022), providing a comprehensive benchmark for evaluating the generalization and robustness. Detailed dataset descriptions are provided in Appendix C.

**Fine-tuning Settings.** To mitigate class imbalance, we construct a balanced training set with 1000 flows per class via sampling or data augmentation. Each dataset is divided into training and testing sets in a ratio of 8:2. Fine-tuning is conducted using the AdamW optimizer with a cosine decay learning rate schedule and a 10% warm-up phase. The learning rates are set to 3e-5 for BERT, 8e-5 for TriFormer to address smaller gradients, and 5e-6 for the Fusion Gate to ensure stable optimization. For each dataset, results are averaged over three runs with different random seeds for stability.

**Evaluation Metrics.** For evaluation, we use four standard classification metrics, *i.e.*, Accuracy (AC), Precision (PR), Recall (RC), and F1-score. We calculate the number of true positives (TP), true negatives (TN), false positives (FP), and false negatives (FN). Based on these, the four metrics are defined as follows:

$$\text{Accuracy} = \frac{TP + TN}{TP + TN + FP + FN}, \text{Precision} = \frac{TP}{TP + FP}. \tag{5}$$

$$\text{Recall} = \frac{TP}{TP + FN}, \text{F1-score} = 2 \times \frac{\text{Precision} \times \text{Recall}}{\text{Precision} + \text{Recall}}. \tag{6}$$

For multi-class tasks, we report macro-averaged Precision, Recall, and F1-score to ensure that each class contributes equally to the overall evaluation. This approach is consistent with our use of balanced sampling and data augmentation to mitigate class imbalance. It provides a comprehensive and fair assessment of both overall performance and per-class performance.

**Baselines.** To comprehensively evaluate the proposed method, we select eight representative baselines, including traditional approaches such as FlowPrint (Van Ede et al., 2020) and AppScanner (Taylor et al., 2016), and deep learning models like FS-Net (Liu et al., 2019a) and GraphDApp (Shen et al., 2021), all relying on spatio-temporal statistical features. We also included four pre-trained models, *i.e.*, ET-BERT (Lin et al., 2022), TrafficFormer (Zhou et al., 2025), NetMamba (Wang et al., 2024), and YaTC (Zhao et al., 2023)—that focus on payload semantics. `TrafficBT` integrates both modalities for enhanced multimodal representation. Comparisons with single-modality models confirm the superiority of our design. All baselines were trained and evaluated on the same datasets for fair comparison. See Appendix D for baseline details.

| Dataset | ISCX-VPN (Service) | | | | ISCX-NonVPN (Service) | | | | ISCX-VPN (App) | | | | ISCX-NonVPN (App) | | | | ISCX-Tor | | | |
|---|---|---|---|---|---|---|---|---|---|---|---|---|---|---|---|---|---|---|---|---|
| Method | AC | PR | RC | F1 | AC | PR | RC | F1 | AC | PR | RC | F1 | AC | PR | RC | F1 | AC | PR | RC | F1 |
| FlowPrint | 0.9728 | 0.9321 | 0.9344 | 0.9292 | 0.8528 | 0.8234 | 0.8276 | 0.8243 | 0.7617 | 0.4388 | 0.5108 | 0.4569 | 0.7895 | 0.6660 | 0.6459 | 0.6366 | 0.7386 | 0.3291 | 0.3452 | 0.3049 |
| AppScanner | 0.6148 | 0.9959 | 0.6148 | 0.7602 | 0.3609 | 0.9819 | 0.3609 | 0.5278 | 0.6825 | 0.9930 | 0.6825 | 0.8090 | 0.3683 | 0.9911 | 0.3683 | 0.5370 | 0.2101 | 0.9681 | 0.2101 | 0.3453 |
| FS-Net | 0.8771 | 0.8998 | 0.8984 | 0.8990 | 0.6492 | 0.7212 | 0.5956 | 0.5957 | 0.9104 | 0.6044 | 0.6305 | 0.6129 | 0.0226 | 0.0017 | 0.0769 | 0.0034 | 0.6944 | 0.3341 | 0.4333 | 0.3770 |
| GraphDApp | 0.5000 | 0.2457 | 0.2262 | 0.1952 | 0.3218 | 0.3573 | 0.2142 | 0.1241 | 0.4793 | 0.0800 | 0.0862 | 0.0618 | 0.4036 | 0.0724 | 0.0793 | 0.0487 | 0.4539 | 0.1900 | 0.2548 | 0.1901 |
| ET-BERT | 0.9375 | 0.9375 | 0.9375 | 0.9375 | 0.7675 | 0.4430 | 0.7630 | 0.5540 | 0.9167 | 0.9167 | 0.9167 | 0.9167 | 0.7153 | 0.7233 | 0.7153 | 0.7173 | 0.5243 | 0.5238 | 0.5238 | 0.5238 |
| TrafficFormer | 0.8784 | 0.8629 | 0.8784 | 0.8669 | 0.7083 | 0.6894 | 0.7083 | 0.6867 | 0.7083 | 0.6894 | 0.7083 | 0.6867 | 0.8333 | 0.8533 | 0.8333 | 0.8339 | 0.5294 | 0.7549 | 0.5284 | 0.5555 |
| NetMamba | 0.9793 | 0.9795 | 0.9793 | 0.9794 | 0.8320 | 0.8315 | 0.8320 | 0.8298 | 0.9056 | 0.9089 | 0.9056 | 0.9056 | 0.9155 | 0.9182 | 0.9155 | 0.9160 | **1.0000** | **1.0000** | **1.0000** | **1.0000** |
| YaTC | 0.9984 | 0.9985 | 0.9984 | 0.9984 | 0.9382 | 0.9385 | 0.9382 | 0.9383 | 0.9762 | 0.9784 | 0.9762 | 0.9763 | 0.9694 | 0.9700 | 0.9694 | 0.9691 | **1.0000** | **1.0000** | **1.0000** | **1.0000** |
| TrafficBT | **1.0000** | **1.0000** | **1.0000** | **1.0000** | **0.9717** | **0.9721** | **0.9717** | **0.9718** | **0.9930** | **0.9931** | **0.9930** | **0.9930** | **0.9830** | **0.9832** | **0.9830** | **0.9829** | **1.0000** | **1.0000** | **1.0000** | **1.0000** |

| Dataset | ISCX-NonTor | | | | CIC-IoT 2022 Attacks(Flood) | | | | USTC-TFC 2016 (Malware) | | | | USTC-TFC 2016 (Benign) | | | | NUDT-Mobile | | | |
|---|---|---|---|---|---|---|---|---|---|---|---|---|---|---|---|---|---|---|---|---|
| Method | AC | PR | RC | F1 | AC | PR | RC | F1 | AC | PR | RC | F1 | AC | PR | RC | F1 | AC | PR | RC | F1 |
| FlowPrint | 0.9597 | 0.8802 | 0.8506 | 0.8613 | 0.7896 | 0.6629 | 0.6078 | 0.5883 | 0.9520 | 0.7466 | 0.8000 | 0.7636 | 0.7365 | 0.7365 | 0.7365 | 0.7365 | 0.4751 | 0.4865 | 0.4509 | 0.4505 |
| AppScanner | 0.5300 | 0.9865 | 0.5300 | 0.6895 | 0.5123 | 0.9957 | 0.5123 | 0.6765 | 0.1702 | 0.9991 | 0.1702 | 0.2908 | 0.5101 | 0.9931 | 0.5101 | 0.6740 | 0.2715 | 0.9910 | 0.2715 | 0.4262 |
| FS-Net | 0.8925 | 0.7702 | 0.5220 | 0.5877 | 0.7674 | 0.6196 | 0.6661 | 0.5849 | 0.8611 | 0.8782 | 0.8349 | 0.8437 | 0.8975 | 0.9481 | 0.9130 | 0.9236 | 0.6375 | 0.6521 | 0.6220 | 0.6272 |
| GraphDApp | 0.4852 | 0.1206 | 0.1452 | 0.0981 | 0.4501 | 0.1622 | 0.2299 | 0.1616 | 0.3660 | 0.4472 | 0.1994 | 0.1831 | 0.5575 | 0.5686 | 0.5262 | 0.4705 | 0.0156 | 0.0001 | 0.0036 | 0.0001 |
| ET-BERT | 0.8318 | 0.8429 | 0.8429 | 0.8429 | 0.4216 | 0.419 | 0.419 | 0.419 | 0.9142 | 0.9142 | 0.9142 | 0.9142 | - | - | - | - | 0.8578 | 0.8618 | 0.8578 | 0.8581 |
| TrafficFormer | 0.7231 | 0.7231 | 0.7231 | 0.7338 | 0.9505 | 0.9534 | 0.9505 | 0.9509 | 0.9677 | 0.9624 | 0.9677 | 0.9636 | 0.6180 | 0.6941 | 0.6180 | 0.6478 | 0.8805 | 0.8844 | 0.8805 | 0.8806 |
| NetMamba | 0.9872 | 0.9873 | 0.9872 | 0.9872 | 0.9980 | 0.9980 | 0.9980 | 0.9980 | 0.9680 | 0.9684 | 0.9680 | 0.9679 | **1.0000** | **1.0000** | **1.0000** | **1.0000** | 0.9329 | 0.9377 | 0.9329 | 0.9333 |
| YaTC | 0.9514 | 0.9499 | 0.9514 | 0.9501 | **1.0000** | **1.0000** | **1.0000** | **1.0000** | 0.9829 | 0.929 | 0.9829 | 0.9829 | **1.0000** | **1.0000** | **1.0000** | **1.0000** | 0.9013 | 0.9502 | 0.9013 | 0.9156 |
| TrafficBT | **0.9942** | **0.9942** | **0.9942** | **0.9942** | **1.0000** | **1.0000** | **1.0000** | **1.0000** | **0.9947** | **0.9949** | **0.9949** | **0.9948** | **1.0000** | **1.0000** | **1.0000** | **1.0000** | **0.9710** | **0.9715** | **0.9710** | **0.9709** |

| Dataset | CrossPlatform (android) | | | | CrossPlatform (ios) | | | | Datacon2020 | | | | Datacon2021 (part1) | | | | Datacon2021 (part2) | | | |
|---|---|---|---|---|---|---|---|---|---|---|---|---|---|---|---|---|---|---|---|---|
| Method | AC | PR | RC | F1 | AC | PR | RC | F1 | AC | PR | RC | F1 | AC | PR | RC | F1 | AC | PR | RC | F1 |
| FlowPrint | 0.8543 | 0.8543 | 0.8543 | 0.8543 | 0.9082 | 0.9082 | 0.9082 | 0.9082 | 0.7250 | 0.8161 | 0.5466 | 0.5052 | 0.2587 | 0.0235 | 0.0909 | 0.0374 | 0.0249 | 0.0002 | 0.0100 | 0.0005 |
| AppScanner | 0.1864 | 0.9794 | 0.1864 | 0.3132 | 0.1314 | 0.9791 | 0.1314 | 0.2318 | 0.6438 | 0.9592 | 0.6438 | 0.7704 | 0.1919 | 0.9809 | 0.1919 | 0.3210 | 0.0732 | 0.9644 | 0.0732 | 0.1361 |
| FS-Net | 0.4614 | 0.2996 | 0.2662 | 0.2710 | 0.3494 | 0.2602 | 0.2420 | 0.2417 | 0.9212 | 0.9177 | 0.8919 | 0.9034 | 0.6971 | 0.8506 | 0.6730 | 0.7221 | 0.0058 | 0.0001 | 0.0100 | 0.0001 |
| GraphDApp | 0.0418 | 0.0002 | 0.0047 | 0.0004 | 0.0486 | 0.0040 | 0.0062 | 0.0019 | 0.6978 | 0.5911 | 0.5075 | 0.4367 | 0.2643 | 0.2707 | 0.1734 | 0.1018 | 0.0252 | 0.0003 | 0.0100 | 0.0005 |
| ET-BERT | 0.9922 | 0.9869 | 0.9783 | 0.9814 | 0.9918 | 0.9831 | 0.9846 | 0.9829 | 0.9550 | 0.9551 | 0.9551 | 0.9550 | 0.7401 | 0.8264 | 0.7203 | 0.7048 | 0.0747 | 0.0194 | 0.0762 | 0.0267 |
| TrafficFormer | 0.7081 | 0.7180 | 0.7087 | 0.7057 | 0.4798 | 0.5141 | 0.4798 | 0.4861 | 0.7708 | 0.7786 | 0.7708 | 0.7692 | 0.9906 | 0.9911 | 0.9906 | 0.9891 | 0.0533 | 0.0260 | 0.0533 | 0.0210 |
| NetMamba | 0.9797 | 0.9817 | 0.9797 | 0.9782 | 0.9837 | 0.9794 | 0.9837 | 0.9807 | 0.8400 | 0.8617 | 0.8400 | 0.8405 | 0.9914 | 0.9926 | 0.9914 | 0.9891 | 0.1134 | 0.1393 | 0.1134 | 0.1008 |
| YaTC | 0.9790 | 0.9792 | 0.9790 | 0.9790 | 0.9084 | 0.9096 | 0.9084 | 0.9082 | 0.9894 | 0.9894 | 0.9894 | 0.9893 | 0.9997 | 0.9997 | 0.9997 | 0.9897 | 0.7814 | 0.7818 | 0.7814 | 0.7800 |
| TrafficBT | **0.9911** | **0.9913** | **0.9911** | **0.9910** | **0.9916** | **0.9901** | **0.9916** | **0.9897** | **0.9947** | **0.9947** | **0.9947** | **0.9947** | **1.0000** | **1.0000** | **1.0000** | **1.0000** | **0.9507** | **0.9505** | **0.9507** | **0.9504** |

Table 4: Performance Comparison on Fifteen Public Datasets Across Eight Baseline Methods. (The best-performing results are highlighted in **Bold**; "-" denotes that the method is not applicable to the dataset.)

## 4.2 Numerical Results

**Downstream Task Classification.** As shown in Table 4, while TrafficBT consistently achieves state-of-the-art results, a detailed analysis reveals specific limitations in baseline models, especially on complex datasets. Traditional methods falter when faced with modern traffic characteristics. FlowPrint, for example, degrades on anonymized and complex traffic, with its F1-score dropping from 0.9292 on ISCX-VPN to 0.3049 on ISCX-Tor. Similarly, AppScanner is biased by class imbalance, showing high Precision but low F1-scores (*e.g.*, 0.5278 on ISCX-NonVPN). Models like FS-Net and GraphDApp demonstrate insufficient feature learning, as they perform poorly on nearly all encrypted and large-scale datasets. More advanced models also exhibit specific weaknesses. ET-BERT and TrafficFormer show performance instability on datasets with bursty patterns or complex port usage, such as CIC-IoT 2022 and NUDT-Mobile. NetMamba, though generally robust, struggles with malicious traffic, showing lower F1-scores on USTC-TFC 2016 (Malware) and Datacon2020. Even the strong baseline YaTC is unstable on tasks with over 100 classes, notably on CrossPlatform (iOS), NUDT-Mobile, and Datacon2021 Part 2, where its F1-score drops to 0.7800. In contrast, TrafficBT overcomes these challenges. By integrating a pre-trained BERT for semantics with a TriFormer for spatio-temporal modeling, it demonstrates superior robustness and generalization. It achieves SOTA results across all tasks, with most scores exceeding 99%, and surpasses YaTC by over 21% on the challenging Datacon2021 Part 2 benchmark.

**Impact of Multimodal Fine-tuning.** To evaluate our multimodal fine-tuning strategy, we conduct an ablation study on the ISCX-NonVPN dataset by training BERT (semantic), TriFormer (spatio-temporal), and the full TrafficBT separately. As shown in Fig. 3b, BERT and TriFormer are pre-trained for 10 and 30 epochs, respectively, before individual fine-tuning. TrafficBT then combines both models for joint fine-tuning. Training and validation performance are measured using

loss and F1-score. Results show that `TrafficBT` achieves lower loss and more stable F1-scores during fine-tuning, demonstrating that multimodal fusion enhances feature learning.

**Impact of Pre-training.** To evaluate the effect of domain-specific pre-training on BERT performance in network traffic modeling, we fine-tune a standard BERT and a domain-pretrained BERT on the ISCX-NonVPN dataset. Loss and F1 score are adopted as evaluation metrics. Fig. 3a shows a 25% F1 drop in standard BERT, demonstrating the benefit of traffic-specific pre-training in modeling network patterns.

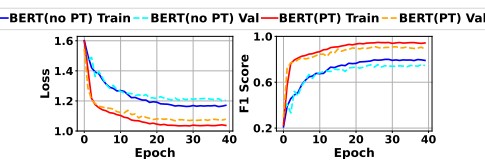
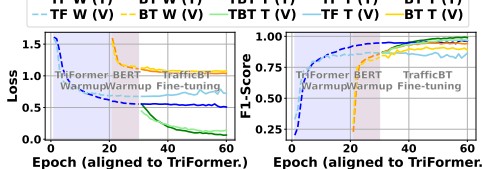

(a) Performance Trends of Domain-Specific Pre-Trained (PT) and Non-Pre-Trained (noPT) BERT Across Fine-Tuning Phases.

(b) Performance Trends of TriFormer (TF), BERT (BT), and `TrafficBT` (TBT) Across Warm-Up and Fine-Tuning Phases.

**Impact of Data Augmentation.** As shown in Table 4, ET-BERT and TrafficFormer perform poorly on ISCX-Tor due to class imbalance. To validate our augmentation strategy, we evaluate loss and F1-score on this dataset. Fig. 4a shows that augmentation significantly improves BERT, TriFormer, and `TrafficBT`, confirming its effectiveness for imbalanced traffic classification. We further assess model robustness on 20 NUDT classes, each containing more than 500 flows, downsampled to 500. Two versions of `TrafficBT`, fine-tuned with and without perturbations within the data augmentation methods, are evaluated under 4 mask probabilities. The augmented model consistently outperforms the non-augmented one in accuracy and F1, demonstrating improved robustness.

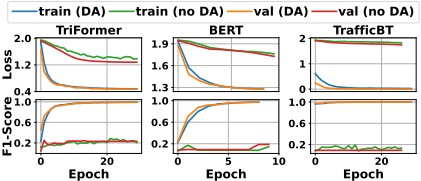
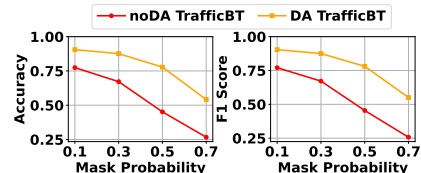

(a) Performance of `TrafficBT` With Data Augmentation (DA) and Without (noDA) During Warm-Up and Fine-Tuning Phases.

(b) Comparison of Model Robustness With Data Augmentation (DA) and Without (noDA).

**Impact of Model Selection in TriFormer.** Given the sequential nature of spatio-temporal features, we compare two lightweight architectures, Transformer and Temporal Convolutional Networks (TCNs) (Bai et al., 2018), for temporal modeling in `TrafficBT`. Both models use the same TriFormer packet-level inputs and are evaluated by training/validation loss and F1 score. Fig. 5 shows that Transformer consistently achieves lower loss and higher F1, proving its superiority as the temporal backbone.

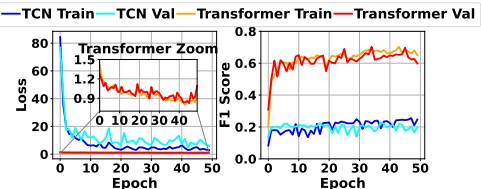

Figure 5: Performance Comparison of TCN and Transformer Models on Spatio-Temporal Features in `TrafficBT`.

## 5 CONCLUSION

This paper introduced `TrafficBT`, a network traffic classification framework that advances pre-trained language models through multimodal representation learning. It achieved state-of-the-art results on fifteen datasets across eight downstream tasks by effectively capturing payload semantics and high-quality spatiotemporal features, outperforming eight existing baselines. Notably, `TrafficBT` also delivered the best performance on the challenging encrypted proxy website classification task, demonstrating the promising potential of pre-trained models in network security management.

ETHICS STATEMENT

The technology, code, and data collection process involved in this manuscript do not involve any ethical risks.

REPRODUCIBILITY STATEMENT

The code, data, and implementation instructions for this paper can be found in the anonymous link `https://anonymous.4open.science/r/TrafficBT-C730`.

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

## USE OF LARGE LANGUAGE MODELS IN MANUSCRIPT PREPARATION

In the preparation of this manuscript, we utilized Large Language Models (LLMs) to aid in polishing the writing. The primary use of these models was to improve the grammar, clarity, and conciseness of the text. All scientific contributions, including the core ideas, experimental design, results, and their interpretation, remain entirely the work of the authors. The LLM served solely as a writing assistance tool.

## APPENDIX

In the supplemental material, we provide the following additional details:

**§A:** We provide a detailed description of the spatio-temporal statistical features utilized in this paper.

   **§A.1:** We describe the 42 flow-level statistical features.

   **§A.2:** We describe the 28 packet-level statistical features.

**§B:** We provide a detailed description of all the datasets utilized for the pre-training phase.

**§C:** We provide a detailed description of all the datasets utilized for the fine-tuning phase across eight downstream tasks.

**§D:** We provide a detailed description of the eight representative baseline models we have experimented with.

**§E:** We provide additional supplementary experiments and results.

   **§E.1:** We present an ablation study comparing three different fusion mechanisms (gating, cross-attention, and residual cross-attention) to validate our model's architectural choice.

## A    SPATIO-TEMPORAL STATISTICAL FEATURES

In this section, we provide a detailed description of the flow-level and packet-level statistical features used in our study, including their definitions, semantic interpretations, and their roles in characterizing network traffic behaviors.

### A.1    FLOW-LEVEL STATISTICAL FEATURES

These network traffic statistical features consist of 42 dimensions across 5 categories, comprehensively characterizing flow behavior from multiple perspectives, as shown in Table 5. A detailed description of each feature category is provided as follows:

**1) Packet Count Statistics**    These features reflect the directionality and frequency of interactions within a flow, which are useful for identifying session patterns or detecting unidirectional anomalies such as DoS attacks.

- *Total Fwd Packets*: The total number of packets transmitted from the source to the destination within a flow.

Table 5: Statistical Features of Network Traffic Flows.

| Category | Feature Name |
|---|---|
| Packet Count Statistics | Total Fwd Packets, Total Bwd Packets |
| Packet Length Statistics | Packet Length Min/Max/Mean/Std/Total (Fwd, Bwd, Flow) |
| Inter Arrival Time (IAT) | IAT Min/Max/Mean/Std/Total (Fwd, Bwd, Flow) |
| TCP Flags | FIN, SYN, RST, PSH, ACK, URG, CWR, ECE Flag Count |
| Flow Rate Statistics | Flow Bytes/s, Flow Packets/s |

- *Total Bwd Packets*: The total number of packets transmitted from the destination back to the source.

**2) Packet Length Statistics**  These statistics capture the payload characteristics of the traffic and are useful for identifying encrypted flows (*e.g.*, fixed-size packets) or abnormal patterns associated with malicious behavior.

- *Packet Length Min/Max/Mean/Std/Total (Fwd, Bwd, Flow)*: The minimum, maximum, average, standard deviation, and total length of packets, computed separately for forward, backward, and overall flow directions.

**3) Temporal Behavior Metrics**  These metrics reflect the temporal dynamics of a flow and help detect anomalies (*e.g.*, high-speed scanning or bot activity) and model application-level behavior.

- *IAT Min/Max/Mean/Std/Total (Fwd, Bwd, Flow)*: The minimum, maximum, average, standard deviation, and sum of inter-arrival times between consecutive packets, computed for forward, backward, and overall directions.

**4) TCP Flag Statistics**  These flags indicate various control and connection states, and unusual flag combinations can reveal potential intrusions such as SYN floods or port scans.

- *FIN, SYN, RST, PSH, ACK, URG, CWR, ECE Flag Count*: The occurrence counts of eight TCP control flags within a flow:
    - *SYN*: connection initiation
    - *FIN*: connection termination
    - *RST*: connection reset
    - *ACK*: acknowledgment
    - *PSH*: push function
    - *URG*: urgent data
    - *CWR*: congestion window reduced
    - *ECE*: explicit congestion notification

**5) Flow Rate Features**  These features are effective for detecting bursty behaviors (*e.g.*, DDoS attacks) and distinguishing between different application types, such as bulk data transfers or real-time services.

- *Flow Bytes/s*: The total number of bytes in the flow divided by the flow duration, representing the data transmission rate.
- *Flow Packets/s*: The total number of packets divided by the flow duration, representing the packet transmission rate.

Table 6: Statistical Features of Network Traffic Packets.

| Category | Feature Name |
|---|---|
| Time-related | timestamp, delta_time, relative_time, time_since_last_handshake |
| Length & Direction | packet_length, payload_length, direction |
| Last-5 Statistics | avg_delta_time_last_5, uplink_ratio_last_5, avg/std_pkt_len_last_5 |
| Protocol & TCP Flags | protocol_id, tcp_flag_(syn/ack/fin), is_ack_only, seq_diff, window_size |
| TLS-related | tls_record_type, tls_version, cipher_suite_len, handshake_phase, key_update_count |
| Content Statistics | entropy, chi_square, printable_ratio, null_byte_ratio, byte_pair_corr |

## A.2 PACKET-LEVEL STATISTICAL FEATURES

These packet-level network traffic statistical features encompass 28 dimensions across 6 categories, providing a comprehensive characterization of packet behavior from diverse aspects, as shown in Table 6.

**1) Time-related Features** These features capture the temporal context of each packet within a flow:

- **timestamp**: The absolute time at which the packet was captured.
- **delta_time**: The time interval between the current packet and the previous packet in the same flow.
- **relative_time**: The elapsed time since the beginning of the flow.
- **time_since_last_handshake**: Time passed since the last observed handshake event (*e.g.*, TLS or TCP), useful for assessing session intervals.

**2) Length and Direction Features** These features describe the size and direction of packet transmission:

- **packet_length**: Total length of the packet, including headers and payload.
- **payload_length**: Length of the payload portion (excluding protocol headers).
- **direction**: Direction of the packet, typically uplink (client to server) or downlink (server to client).

**3) Last-5 Statistics** These short-term statistics summarize recent flow behavior based on the last five packets:

- **avg_delta_time_last_5**: Average inter-arrival time over the last five packets.
- **uplink_ratio_last_5**: Proportion of uplink packets among the last five.
- **avg_pkt_len_last_5**, **std_pkt_len_last_5**: Average and standard deviation of packet lengths over the last five packets.

**4) Protocol and TCP Flag Features** These features capture transport-layer protocol behaviors and TCP control signals:

- **protocol_id**: Numerical identifier for the protocol (*e.g.*, TCP, UDP).
- **tcp_flag_syn**, **tcp_flag_ack**, **tcp_flag_fin**: Indicators for the presence of SYN, ACK, and FIN flags.

- **is_ack_only**: Flag indicating whether the packet contains only an ACK without payload or other flags.
- **seq_diff**: Difference in TCP sequence numbers between consecutive packets.
- **window_size**: Advertised TCP window size reflecting buffer capacity.

**5) TLS-related Features**   These features describe encrypted session characteristics:

- **tls_record_type**: Type of TLS record (*e.g.*, handshake, application data).
- **tls_version**: TLS protocol version used.
- **cipher_suite_len**: Length of the cipher suite list.
- **handshake_phase**: Current phase of the TLS handshake process.
- **key_update_count**: Number of observed TLS key update events.

**6) Content Statistics**   These features evaluate the statistical properties of packet payloads:

- **entropy**: Randomness or unpredictability in packet content.
- **chi_square**: Deviation of byte frequency from a uniform distribution.
- **printable_ratio**: Ratio of printable ASCII characters in the payload.
- **null_byte_ratio**: Proportion of null (zero) bytes, indicating binary or compressed data.
- **byte_pair_corr**: Correlation between adjacent byte pairs to assess structural redundancy.

## B   PRE-TRAINING DATASET

In this section, we provide a detailed introduction to the datasets used for pre-training in this study, as shown in Table 7. For model pre-training, we employed four publicly available datasets: NUDT-Mobile, ISCX-VPN-NonVPN, ISCXTor2016, and CIRA-CIC-DoHBrw-2020. These datasets collectively offer a broad and diverse representation of network traffic scenarios. Among them, NUDT-Mobile is a recently collected large-scale real-world mobile traffic dataset from the National University of Defense Technology (NUDT), which is incorporated to enhance the model's robustness in diverse and realistic mobile network environments. A detailed description of each dataset is provided below:

Table 7: Overview of Pre-training Datasets.

| Dataset | Size | #Flows | #Classes | Included Protocols |
|---------|------|--------|----------|--------------------|
| NUDT-Mobile | 112.2 GB | 1,157,245 | 280 | TCP, UDP, HTTP, TLSv1.2, SSLv2, WebSocket, ... |
| ISCXVPN2016 | 15.6 GB | 4,824 | 5 | TLSv1.2, SFTP, SSDP, SNMP, NTP, GQUIC, ... |
| ISCXTor2016 | 19.7 GB | 39,018 | 7 | TLSv1.1, TLSv1.2, FTP-DATA, SSL, HTTP, WebSocket, ... |
| CIRA-CIC-DoHBrw-2020 | 75.5 GB | 771,497 | 2 | TCP, TLSv1.2, TLSv1.3, SSLv2, SSL, ... |

**NUDT-Mobile Dataset (Zhao et al., 2024).**   This dataset is a large-scale, real-world Android traffic dataset collected by the National University of Defense Technology to address the lack of accurately labeled and shareable mobile application traffic. Collected between May and July 2020 using a custom framework based on Android's `VPNService` and NetLog, it contains 611.23 GB of labeled traffic generated by 224 volunteers using 94 Android devices (Android 6–10) across 9 brands. Network types include WiFi (85%) and mobile (3G/4G/LTE), and 350 mainstream apps from 22 categories are covered, each contributing at least 100MB of traffic.

The released dataset includes anonymized `pcap` files and corresponding label logs, along with supplemental files describing application metadata, device information, and byte distribution statistics. Owing to its scale, labeling precision, and thorough anonymization, NUDT-Mobile serves as a strong benchmark for mobile traffic analysis and encrypted application classification.

**ISCXVPN2016 Dataset (Gil et al., 2016).**   This dataset provides a diverse and well-labeled collection of real-world network traffic for evaluating encrypted traffic analysis. It includes flows from various application categories under both regular and VPN (OpenVPN/UDP) conditions, enabling fine-grained classification and behavioral comparisons. Researchers simulated user behavior using two accounts (Alice and Bob) interacting with applications like Skype, Facebook, Gmail, and uTorrent. Traffic was captured in controlled environments using *Wireshark* and *tcpdump*, with only the target app active during each session to ensure accurate labeling.

The dataset spans 14 traffic categories, each with both VPN and non-VPN variants, including web browsing, email, chat, streaming, file transfer, VoIP, and P2P. It provides full packet captures (`.pcap`) and flow-level summaries (`.csv`) generated via ISCXFlowMeter. Thanks to its realistic traffic structure and rich labels, this dataset is widely used for VPN detection, encrypted traffic classification, and intrusion detection benchmarking.

**ISCXTor2016 Dataset (Lashkari et al., 2017).**   This dataset is a labeled network traffic dataset collected by the Canadian Institute for Cybersecurity (CIC) to support anonymized traffic analysis. It contains both Tor-based and regular traffic flows across a wide range of applications, enabling comparison between encrypted and non-encrypted behaviors. Five simulated users generated traffic from popular services, including browsing (Chrome, Firefox), chat (Skype, Facebook, Hangouts), email (Thunderbird), file sharing (BitTorrent), and multimedia streaming (Spotify, YouTube, Vimeo).

Tor-based traffic spans eight representative categories such as browsing, email, chat, streaming, file transfer, VoIP, and P2P. Traffic was captured simultaneously at both the user end and a gateway node to align Tor and non-Tor sessions, with labels verified based on application usage. The dataset includes full packet captures (`.pcap`) and flow-level summaries (`.csv`) generated via `ISCXFlowMeter`, making it a valuable benchmark for anonymized traffic classification and privacy-preserving service analysis.

**CIRA-CIC-DoHBrw-2020 Dataset (MontazeriShatoori et al., 2020).**   This dataset was released by the Canadian Institute for Cybersecurity in collaboration with CIRA to support DNS over HTTPS (DoH) traffic analysis.  It contains labeled traffic categorized into non-DoH, benign-DoH, and malicious-DoH flows.  Data was collected using five web browsers and multiple DoH resolvers (AdGuard, Cloudflare, Google, and Quad9), with malicious flows generated by tunneling tools such as dns2tcp, DNSCat2, and Iodine to simulate covert channels.

To enhance flow-level analysis, the dataset introduces the concept of packet clumps—aggregated sequences of packets in the same direction—reducing noise from control packets. It includes full packet captures (`.pcap`) and flow-level features extracted by `DoHLyzer` and `DoHMeter`, covering 28 statistical and temporal attributes. This dataset is widely used for DoH traffic detection, DNS tunneling analysis, and encrypted communication threat modeling.

**Pre-training Dataset Summary**   By integrating diverse datasets, the pre-training stage benefits from broad exposure to encrypted and anonymized traffic, enhancing the robustness and generalization of learned representations. This enables effective performance in downstream tasks such as traffic classification, anomaly detection, and privacy-aware analysis.

## C   FINE-TUNING DATASET

This section provides a detailed overview of the 15 fine-tuning datasets used in this study, including their sources, traffic categories, file formats, and other relevant details, as shown in Table 8. To evaluate the generalization capability of the proposed model, we utilize 15 publicly available datasets spanning eight classification tasks. These datasets cover a wide range of real-world network traffic scenarios, serving as comprehensive benchmarks for downstream evaluation. A brief summary of each dataset and its application context is presented below.

Table 8: Overview of Fine-tuning Datasets.

| Task | Dataset | #Flows | #Classes |
|---|---|---|---|
| VPN Traffic Classification | ISCX-VPN (Service, App) | 1,816 / 3,008 | 5 / 5 |
| Tor Traffic Classification | ISCX-Tor | 172 | 7 |
| Service Classification | ISCX-NonVPN (Service), ISCX-NonTor | 3,008 / 38,846 | 5 / 7 |
| Application Classification | ISCX-NonVPN (App), USTC-TFC (Benign), CrossPlatform (Android, iOS), NUDT-Mobile | 3,159 / 231,449 / 114,824 / 46,829 / 24,529 | 13 / 280 / 10 / 211 / 195 |
| Malware Classification | USTC-TFC (Malware), Datacon2020 | 92,587 / 63,980 | 10 / 2 |
| Proxy Classification | Datacon2021 (Part 1) | 21,005 | 11 |
| Website Classification (Proxy) | Datacon2021 (Part 2) | 32,516 | 100 |
| Device Classification (Under Attacks) | CIC-IoT 2022 (Flood) | 27963 | 100 |

**ISCXVPN2016(Service, App) (Gil et al., 2016).** This dataset includes labeled VPN and non-VPN traffic with service and application annotations. A comprehensive overview is provided in the section B.

**ISCXTor2016 (Lashkari et al., 2017).** This dataset captures traffic in both Tor and non-Tor environments, facilitating research on anonymous and obfuscated traffic detection. Refer to the pre-training dataset descriptions in the section B for further details.

**USTC-TFC-2016 (Benign, Malware) (Wang et al., 2017).** This dataset is a publicly available dataset for malware traffic classification, addressing issues like limited data volume and lack of raw traffic in earlier studies. It contains 3.71 GB of raw PCAP files split into 20 categories—10 malicious and 10 normal. Malicious traffic was sourced from CTU malware traces (2011–2015), while normal traffic was generated using IXIA BPS, covering typical applications like P2P, VoIP, email, social media, and gaming.

A dedicated tool, USTC-TK2016, converts PCAPs into deep learning–ready image-like samples, resulting in 752,000 labeled flows. The dataset supports both flow-level and payload-level analysis, making it a valuable benchmark for traffic classification using machine learning and deep learning techniques.

**CrossPlatform (Android, iOS) (Ren et al., 2019).** This dataset contains labeled network traffic collected from 600 popular Android and iOS apps across China, the US, and India, with data captured between August and November 2017 on iOS 10 and Android 6 devices. Each app was manually interacted with for five minutes, and traffic was intercepted using Mitmproxy to analyze HTTP/HTTPS transmissions. PII leakage was detected using the ReCon machine learning tool.

The dataset records detailed PII types (*e.g.*, IMEI, GPS, email), encryption status (plaintext vs. encrypted), communication protocols, recipient types (first-party vs. third-party), and contacted domains (*e.g.*, Google, Umeng). Statistical comparisons reveal cross-national differences in privacy exposure, such as higher PII leakage in Indian apps and more plaintext transmission in Chinese apps. The dataset includes raw traffic traces and annotated summaries, serving as a valuable benchmark for mobile app privacy analysis under varying regulatory and cultural conditions.

The dataset used in our experiments was publicly available at the time of experimentation and submission. However, the hosting site is currently inaccessible, and the dataset is no longer publicly available.

**NUDT-Mobile (Zhao et al., 2024).** A recently collected large-scale real-world mobile network dataset based on Android devices. It covers diverse application types and user behaviors, aiding in enhancing the robustness and generalization of mobile traffic classification models. For detailed

dataset description and collection methodology, please refer to the specific introduction in the section B.

**Datacon2020 (DataCon Community, 2021a).** This dataset was provided by QiAnXin Technology Research Institute to support research on encrypted malware traffic detection. It contains TLS/SSL-encrypted network traffic generated by running malware and benign Windows executables in a controlled sandbox environment between February and June 2020. All samples communicate over port 443.

The dataset includes 6,000 training samples (3,000 malware and 3,000 benign) and 4,000 test samples (2,000 each), with a clear temporal split: malicious training samples are from Feb–May 2020, and test samples from June 2020. All benign samples were collected throughout 2020. This dataset provides a realistic and balanced benchmark for evaluating encrypted traffic classification methods under evolving behavioral and temporal conditions.

**Datacon2021 (Part 1 & 2)** This dataset captures extensive encrypted network traffic generated by various proxy software and different websites accessed through these proxies. It is designed to support two core research tasks: identifying the proxy software based on traffic characteristics and recognizing websites accessed through encrypted proxies.

In **Part 1**, the focus is on classifying traffic according to the proxy software that generated it. The dataset provides labeled PCAP samples organized in a `sample` folder, where filenames encode the proxy category and sample index (*e.g.*, `label_n.pcap`). Additionally, a `real_data` folder contains unlabeled traffic from diverse proxy software, allowing for evaluation and testing. The proxy software categories include popular tools such as OpenVPN (UDP and TLS versions), Psiphon (TCP and TLS), V2Ray, Clash, Lantern, WireGuard, Shadowsocks, Firefox, and others.

Traffic was generated in a controlled Windows environment using automated browsing scripts with Python's `selenium` to visit curated website lists, while `tshark` was employed for traffic capture.

**Part 2** addresses the task of identifying websites based on encrypted traffic routed through a single proxy software. This portion contains training data with labeled PCAP files in a `train_data` folder, and a `test_data` folder with unlabeled traffic samples generated by different websites through the same encrypted proxy. This setup supports research in website fingerprinting and encrypted traffic classification under proxy obfuscation.

Overall, this dataset provides a rich benchmark for studying encrypted proxy traffic behavior, enabling the development of models for proxy identification and website classification in encrypted network environments.

**CIC-IoT 2022(Flood) (Dadkhah et al., 2022).** This dataset contains labeled network traffic from a variety of IoT devices communicating over Wi-Fi (IEEE 802.11), Zigbee, and Z-Wave, captured in a controlled lab setting. It supports IoT device profiling, behavior modeling, and attack detection across six experiment types: *Power*, *Idle*, *Interactions*, *Scenarios*, *Active*, and *Attacks*.

Traffic was captured using Wireshark and dumpcap from dual-interface machines connected to both the gateway and local IoT devices via unmanaged switches. Attack scenarios include RTSP brute force and flooding attacks using tools like Nmap and Hydra. The dataset includes packet capture files (`.pcap`) and flow-level statistics for each device or scenario, making it a valuable benchmark for IoT traffic classification, anomaly detection, and security research in smart environments.

**Fine-tuning Dataset Summary** The datasets employed in this study span a wide range of network traffic types, including VPN, proxy, malware, mobile, IoT, and DNS over HTTPS (DoH) traffic. They collectively cover diverse benign and malicious behaviors across various network environments and device categories. With rich annotations and realistic traffic patterns, these datasets enable robust training and evaluation for tasks such as encrypted traffic classification, anomaly detection, device profiling, and privacy risk analysis. Their diversity ensures comprehensive assessment of model generalization and effectiveness in real-world network security applications.

## D BASELINES

To facilitate reproducibility and provide further insights, we briefly describe the implementation details and core characteristics of the eight representative baselines evaluated in this study.

**FlowPrint (Van Ede et al., 2020).** FlowPrint is a fingerprinting-based method for encrypted mobile traffic classification, capable of identifying known applications and detecting previously unseen ones without relying on payload content or prior knowledge. It operates through five key steps: feature extraction from TCP/UDP flows (*e.g.*, destination IP/port, TLS certificate metadata, timing, packet sizes), destination-based clustering, browser traffic filtering via Random Forest, construction of a temporal co-occurrence graph, and fingerprint extraction using maximal cliques matched by Jaccard similarity.

FlowPrint leverages rich metadata features, including destination attributes (IP, port, TLS fields), temporal patterns (inter-flow intervals, packet timings), and basic size statistics. It processes .pcap files directly and is well-suited for TLS-encrypted environments. Default parameters include a 300s batching window, 30s co-occurrence window, and a 0.9 similarity threshold. The method has been evaluated on multiple datasets such as ReCon, Cross-Platform, Andrubis, and Browser, demonstrating its effectiveness in practical encrypted traffic scenarios.

**AppScanner (Taylor et al., 2016).** AppScanner is a lightweight baseline for Android app identification under encrypted traffic (*e.g.*, HTTPS/TLS), relying solely on flow-level statistical features without decrypting payloads or using IP/DNS information. It collects labeled traffic via ADB and UI automation tools by launching one app at a time, preprocesses .pcap files into flows segmented by bursts and ports, and extracts 40 statistical features (*e.g.*, min/max/mean size, quantiles, packet count, skewness) computed separately for inbound, outbound, and bidirectional directions.

A supervised classifier (typically Random Forest with 150 trees) is trained using one of three strategies: Per Flow Length, Single Large Classifier, or Per App Classifier. The best performance is achieved by the Per App strategy with a confidence threshold of 0.7, yielding over 99% accuracy across 110 Google Play apps. AppScanner supports both online and offline identification modes, and uses real-time rejection of low-confidence predictions to reduce false positives.

**FS-Net (Liu et al., 2019a).** FS-Net is an end-to-end deep learning model for encrypted traffic classification that requires no manual feature engineering. It takes raw packet length sequences as input and directly predicts application labels. The architecture features embedding layers, bidirectional GRU-based encoder and decoder modules, a reconstruction mechanism to enhance representation learning, and an MLP-based classifier.

FS-Net supports packet length sequences by default, with optional integration of message type sequences. Key strengths include end-to-end learning, auxiliary reconstruction loss, lightweight design, and extensibility. On a benchmark of 18 application classes, it achieves 99.14% true positive rate and 0.9906 accuracy, outperforming traditional sequence-based models.

**GraphDApp (Shen et al., 2021).** GraphDApp is a GNN-based model for encrypted DApp traffic classification in the Ethereum ecosystem. It represents each traffic flow as a Traffic Interaction Graph (TIG), where packets are nodes and edges encode temporal and burst relationships. No manual features are engineered; packet lengths and directions serve as node attributes.

The model uses three graph convolutional layers with ReLU and dropout, sum pooling for node aggregation, and an MLP classifier. Trained with Adam optimizer, GraphDApp achieves 89.22% accuracy on 40-class closed-world classification and 99.73% AUC in open-world scenarios with 1,260 background apps. It also generalizes well to mobile app traffic. In summary, GraphDApp leverages graph structures and GNN embeddings to provide a robust, end-to-end baseline for encrypted traffic classification without handcrafted features.

**ET-BERT (Lin et al., 2022).** ET-BERT is a pre-trained Transformer model for encrypted traffic classification that removes the need for handcrafted features. It learns contextual byte-level representations from over 30GB of unlabeled encrypted traffic via self-supervised pre-training, then fine-tunes on various downstream tasks including application identification, VPN detection, and

TLS 1.3 classification. The input encodes raw encrypted byte streams into directional BURST token sequences using a combination of bi-gram and Byte-Pair Encoding. Pre-training uses two objectives: Masked BURST Modeling (predict masked tokens) and Same-origin BURST Prediction (determine if sub-BURST pairs come from the same burst) to capture local and transmission semantics.

ET-BERT uses a 12-layer Transformer with BERT-base configurations and supports packet- or flow-level inputs during fine-tuning. It achieves state-of-the-art results on multiple benchmarks, *e.g.*, 92.5% F1 on Cross-Platform, 98.9% on ISCX-VPN-Service, and 97.41% on CSTNET-TLS 1.3, and shows strong few-shot performance even with limited labeled data. In summary, ET-BERT provides a robust and versatile baseline for encrypted traffic analysis by leveraging burst-aware tokenization and self-supervised learning to capture rich byte-level semantics.

**TrafficFormer (Zhou et al., 2025).** TrafficFormer is a Transformer-based model for encrypted traffic classification that leverages burst-level encoding and a hybrid pre-training framework. It employs two pre-training tasks—Masked Burst Modeling (MBM) to learn local contextual semantics by predicting masked tokens within bursts, and Same-Origin-Direction-Flow (SODF) classification to capture directional and flow-level structural dependencies. During fine-tuning, Random Initialization Field Augmentation (RIFA) randomly resets irrelevant header fields (*e.g.*, IPID, TCP sequence number) to enhance model robustness against protocol variations.

TrafficFormer's input is constructed by extracting 64 bytes from the first five packets per flow, encoding them as hexadecimal bigrams, applying Byte Pair Encoding (BPE), and formatting into sequences compatible with a 12-layer BERT-base Transformer. Pre-trained on large-scale, unlabeled encrypted traffic, TrafficFormer achieves strong performance on downstream tasks such as application identification and protocol detection by effectively modeling burst semantics and flow structure with augmented robustness.

**NetMamba (Wang et al., 2024).** NetMamba is a lightweight baseline for encrypted traffic classification based on a Mamba architecture that replaces Transformer self-attention with a linear-complexity state space model (SSM) for efficient long-range dependency modeling. It represents each flow using the first five packets, extracting 80 bytes of header and 240 bytes of payload per packet, forming a 1600-byte sequence divided into 401 non-overlapping 4-byte strides. The model is pre-trained via masked autoencoding, reconstructing 90% randomly masked strides, and fine-tuned using a simple MLP classifier on the encoder's `[CLS]` token. Input tokens are embedded directly without vocabulary or BPE tokenization, with learnable positional embeddings preserving order.

NetMamba's flow segmentation removes sensitive identifiers to prevent shortcut learning, making it suitable for large-scale or latency-sensitive deployment. In summary, NetMamba offers an efficient, real-time-capable baseline that leverages stride-based input and linear SSM modeling without relying on CNNs or handcrafted features.

**YaTC (Zhao et al., 2023).** YaTC (Yet Another Traffic Classifier) is a strong baseline for encrypted traffic classification in low-resource scenarios. It introduces a Multi-Level Flow Representation (MFR) that converts each flow's first five packets into a 40×40 byte matrix, preserving multi-level semantics while mitigating dominance from long packets. This matrix is split into 400 non-overlapping 2×2 patches, each flattened and linearly embedded into 192-dimensional tokens with positional encoding. YaTC employs a ViT-style Transformer encoder pre-trained using a masked autoencoder (MAE) strategy, masking 90% of patches and reconstructing them with MSE loss to learn robust byte-level features. For downstream tasks, the decoder is removed, and the encoder output undergoes row- and column-wise average pooling before classification via an MLP.

Flows are segmented by 5-tuple keys, with anonymization masking MAC, IP, and port fields to avoid shortcuts. Packets are flattened into 320-byte vectors forming the input matrix. In summary, YaTC offers a compact, expressive model that captures structural flow semantics without payload inspection, delivering strong performance especially in few-shot and encrypted traffic classification tasks.

**Baseline Summary** The first four baselines (FlowPrint, AppScanner, FS-Net, and GraphDApp) rely on various hand-crafted or learned representations of spatio-temporal statistical features, all

of which are fully covered by the statistical features extracted by `TrafficFBERT`, facilitating a fair and consistent comparison. The remaining four models (ET-BERT, TrafficFormer, NetMamba, and YaTC) leverage pre-trained architectures with payload-level inputs, offering strong semantic baselines for benchmarking multimodal fusion performance.

# E    SUPPLEMENTARY EXPERIMENTS

## E.1    FUSION MODEL SELECTION

To validate the design choice of the fusion module in `TrafficBT`, we conducted supplementary experiments by comparing three variants: (1) cross-attention, (2) cross-attention with residual connections, and (3) a gating mechanism. The comparisons were performed in both the warming-up and finetuning stages, with the F1-score used as the evaluation metric.

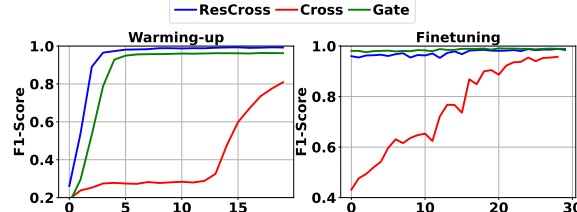

Figure 6: Performance Comparison of Fusion Module Variants in `TrafficBT` During Warm-up and Fine-tuning Stages (F1-Score). ResCross denotes cross-attention with residual connections, Cross denotes plain cross-attention, and Gate denotes the gating mechanism.

As shown in Fig. 6, during the warming-up stage, the gating mechanism achieves slightly lower F1-scores than cross-attention with residual connections, while plain cross-attention performs the worst and converges much more slowly. In the finetuning stage, the gating mechanism and cross-attention with residual connections yield comparable F1-scores, whereas cross-attention remains significantly inferior in both performance and convergence speed. Considering that cross-attention introduces higher computational cost and greater architectural complexity, we ultimately adopt the simpler gating mechanism as the fusion module of `TrafficBT`.

