# OpenReview forum: "TrafficBT: Advancing Pre-trained Language Models for Network Traffic  Classification with Multimodal Traffic Representations"
_ICLR.cc/2026/Conference — ICLR 2026 Conference Withdrawn Submission_

### Official Review · Reviewer_hLDJ · 2025-10-29

**Soundness:** 2
**Presentation:** 3
**Contribution:** 2
**Rating:** 2
**Confidence:** 3

**Summary:**

The paper proposes TrafficBT, a multimodal pre-training and fine-tuning framework for encrypted traffic classification. It integrates a BERT encoder with a lightweight Transformer module (TriFormer) to model payload semantics and spatio-temporal statistical features, which are fused through a gating network. In addition, the framework employs data augmentation strategies—including masking, shuffling, and noise perturbation—to address class imbalance and enhance model robustness.

**Strengths:**

1.	Clear motivation: The paper identifies a meaningful gap in existing pre-trained traffic models that mainly capture payload semantics, and systematically extends this paradigm to multimodal (semantic + spatio-temporal) learning.

2.	Well-written and well-organized: The paper is clearly structured and effectively presented, with smooth transitions and clear figures, which greatly improve readability.

3.	Extensive experiments on 15 datasets: The authors conduct a comprehensive evaluation on 15 public datasets covering eight typical downstream tasks, including VPN detection, malware classification, encrypted proxy identification, and cross-platform traffic analysis.

**Weaknesses:**

1.	Incremental innovation: The novelty is limited. Existing approaches such as ET-BERT, TrafficFormer, and YaTC already employ pre-training for traffic classification. The contribution mainly extends these methods by adding a spatio-temporal encoding module (TriFormer) and a fusion gate, which appears as an incremental improvement rather than a conceptual breakthrough.

2.	Insufficient experimental validation: Although the paper compares TriFormer with a TCN-based temporal encoder (Figure 5), it still lacks ablation experiments isolating the effect of the spatio-temporal module itself. Specifically, results without any spatio-temporal encoding (e.g., using only the BERT backbone) are not presented, making it difficult to quantify the overall contribution of TriFormer to performance gains.

3.	Unclear dataset setup and potential data leakage: Although the paper mentions the use of a “leave-one-out” strategy, the datasets employed for pre-training (e.g., NUDT-Mobile, ISCXVPN2016, Tor2016) appear to share similar distributions and application types with those used for fine-tuning. The authors should provide more detailed descriptions of the datasets to demonstrate that there is no significant risk of feature or distribution leakage, thereby ensuring that the reported performance can be fairly evaluated.

**Questions:**

1.	Clarification on Bigram Splitting Strategy: In Lines 177–179, the payload feature is extracted by splitting a sequence into overlapping bigrams; could the authors elaborate on the motivation and advantages of this design choice?

2.	Limited modeling of spatio-temporal dependencies: The current spatio-temporal representations seem to rely mainly on simple flow- and packet-level statistics, rather than capturing fine-grained inter-packet or intra-packet relationships. It remains unclear whether the proposed Packet Feature Meta-Sequence Learning can effectively model such complex dependencies. Additional clarification and discussion on this aspect would strengthen the paper.

3.	Missing strong baselines: The paper should compare against more recent and advanced baselines such as TrafficLLM[1] to provide a fairer performance benchmark.

4.	Scalability and efficiency: How do the computational costs of TrafficBT compare with models like NetMamba or YaTC? Since the proposed framework combines BERT, Transformer, and a gating network, it is unclear whether it remains computationally efficient.

5.	Lack of hyperparameter sensitivity analysis: Most hyperparameter choices appear to be empirical, and no sensitivity analysis is provided. For example, the 10% masking, shuffling, and noise perturbation rate used to simulate real-world network noise lacks experimental justification. The authors are encouraged to conduct hyperparameter experiments to evaluate model robustness and provide stronger support for these design choices.


[1] Tianyu Cui, et al. TrafficLLM: Enhancing Large Language Models for Network Traffic Analysis with Generic Traffic Representation. arXiv

---

### Official Review · Reviewer_bQYM · 2025-10-29

**Soundness:** 2
**Presentation:** 2
**Contribution:** 2
**Rating:** 4
**Confidence:** 3

**Summary:**

The paper introduces TrafficBT, a framework for overcoming the limitations of existing pre-trained models in network traffic classification. In general, a network traffic classification task maily depends on single-modality payload semantics. The introduced approach utilizes a multimodal concept with consideration of packet payload information with additional modalities (temporal dependencies between packets for making traffic representation). The model employs a pre-trained backbone for semantic feature extraction with a gating mechanism for multimodal fusion.

**Strengths:**

S1) The authors attempts to address the current limitation in NTC tasks by introducing temporal and potentially static features is a significant step towards creating a truly robust traffic fingerprint.

S2) Utilizing the gating mechanism allows to address the task with a commitment to its efficiency.

S3) By basing the semantic stream on established pre-trained language models, the framework benefits from deep linguistic understanding of the payload, providing a powerful, high-quality feature backbone before fusion occurs.

**Weaknesses:**

W1) The authors do not consider any detailed quantitative approaches for the contribution of each modality. It means that it is insufficient for showing how much of the performance is provided from the temporal features.

W2) Selecting the gating mechanism may limit the complexity of the interactions, which can be trained between modalities.

W3) The robustness and generalization issues should be considered and validated acrosss a wider range of heterogeneous and noisy datasets.

**Questions:**

Q1) Can we meet the dimensionality of the modality embeddings before the fusion layer?

Q2) Can we review any metrics for computational costs of the employed approaches and modules?

Q3) Do the authors considered early fusion (concatenating raw inputs or early embeddings) or late fusion (fusing only the classification logits)?

Q4) Is there any specific information for the learned gate weights?

---

### Official Review · Reviewer_uWj7 · 2025-11-01

**Soundness:** 2
**Presentation:** 2
**Contribution:** 2
**Rating:** 4
**Confidence:** 4

**Summary:**

This paper proposed TrafficBT, which combines BERT with Transformer to model spatio-temporal characteristics of network traffic and achieve multimodal fusion. It also introduced the Gating Network to adaptively combine payload semantics and transmission modes to improve the generalization ability of the model in complex scenarios.

**Strengths:**

+ Systematic evaluation was carried out on several public datasets, covering diversity types of downstream tasks. The experimental scale is large.
+ Design modal-specific data enhancement strategy for category imbalance and sample scarcity in traffic data to effectively improve the robustness of the model.
+ Provides codes to enhance the reproducibility and impact of research.

**Weaknesses:**

+ The pre-trained datasets and fine-tuned data overlap, and the results on the NUDT-Mobile, ISCXVPN2016 and ISCXTor2016 data sets may be overestimated, which raises concerns about generalization.
+ The methods used are a combination of existing methods, such as BERT, data enhancement and Transformer, which have been included in previous work respectively. The technical novelty of TrafficBT is limited.
+ Traffic BT has a complex structure and includes multiple modules such as BERT, TriFormer, and gated Network. Although it performed well in experiments, its deployment feasibility in resource-limited scenarios (such as edge devices, real-time detection) has not been fully evaluated.
+ In this paper, payload semantics and statistical characteristics are regarded as "multimodal", but both essentially come from the same traffic data. It is recommended to define the concept of "multimodal" more strictly, or consider introducing truly heterogeneous modes (such as traffic + log, etc.).
+ There are some errors in presentation, such as the missing captions of Figure 3 and Figure 4.

**Questions:**

+ Why choose 80% as the MASK rate during pre-training?
+ In Fusion, how is parameter a selected, and how to identify the importance of features extracted by BERT and TriFormer for different datasets?
+ Why BERT can capture semantic information.
+ Could you elaborate on how the training set and test set are segmented during data processing, whether based on flow, packet or random? In addition, the overview of the test dataset is missing, and how you avoid test data leakage?
+ Will you release the pre-trained model and the processed datasets to promote the related research?

---

### Note · Authors · 2025-11-23

I have read and agree with the venue's withdrawal policy on behalf of myself and my co-authors.